# Isolation of Human Lineage, Fluoroquinolone-Resistant and Extended-β-Lactamase-Producing *Escherichia coli* Isolates from Companion Animals in Japan

**DOI:** 10.3390/antibiotics10121463

**Published:** 2021-11-28

**Authors:** Toyotaka Sato, Shin-ichi Yokota, Tooru Tachibana, Satoshi Tamai, Shigeki Maetani, Yutaka Tamura, Motohiro Horiuchi

**Affiliations:** 1Laboratory of Veterinary Hygiene, Faculty of Veterinary Medicine, Hokkaido University, Sapporo 060-0818, Japan; horiuchi@vetmed.hokudai.ac.jp; 2Graduate School of Infectious Diseases, Hokkaido University, Sapporo 060-0818, Japan; 3Department of Microbiology, Sapporo Medical University School of Medicine, Sapporo 060-8556, Japan; syokota@sapmed.ac.jp; 4Hokko Animal Hospital, Sapporo 065-0014, Japan; ceo@hokkou-ac.jp; 5Tamai Animal Hospital, Sapporo 062-0034, Japan; vet_tamai@me.com; 6Maetani Animal Hospital, Sapporo 002-8003, Japan; mae-hos@sa2.so-net.ne.jp; 7School of Veterinary Medicine, Rakuno Gakuen University, Professor Emeritus, Ebetsu 069-8501, Japan; yutatamu918@gmail.com

**Keywords:** fluoroquinolone-resistance, ESBL, *Escherichia coli*, dogs, cats

## Abstract

An increase in human and veterinary fluoroquinolone-resistant *Escherichia coli* is a global concern. In this study, we isolated fluoroquinolone-resistant *E. coli* isolates from companion animals and characterized them using molecular epidemiological analysis, multiplex polymerase chain reaction to detect *E. coli* ST131 and CTX-M type extended-spectrum β-lactamases (ESBL), and multi-locus sequence typing analysis. Using plain-CHROMagar ECC, 101 *E. coli* isolates were isolated from 34 rectal swabs of dogs and cats. The prevalence of resistance to fluoroquinolone and cefotaxime was 27.7% and 24.8%, respectively. The prevalence of fluoroquinolone-resistant isolates (89.3%) was higher when CHROMagar ECC with CHROMagar ESBL supplement was used for *E. coli* isolation. The prevalence of cefotaxime resistance was also higher (76.1%) when 1 mg/L of ciprofloxacin-containing CHROMagar ECC was used for isolation. The cefotaxime-resistant isolates possessed CTX-M type β-lactamase genes (CTX-M-14, CTX-M-15, or CTX-M-27). Seventy-five percent of fluoroquinolone-resistant isolates were sequence types ST131, ST10, ST1193, ST38, or ST648, which are associated with extensive spread in human clinical settings. In addition, we isolated three common fluoroquinolone-resistant *E. coli* lineages (ST131 clade C1-M-27, C1-nM27 and ST2380) from dogs and their respective owners. These observations suggest that companion animals can harbor fluoroquinolone-resistant and/or ESBL-producing *E. coli*, in their rectums, and that transmission of these isolates to their owners can occur.

## 1. Introduction

For many people, companion animals such as dogs and cats are a part of their daily lives. These animals may be given antimicrobial agents to treat bacterial infections, such as urinary tract infections, pneumonia, and blood stream infections [1,2,3].

Antimicrobial resistance is of a major concern worldwide. In Japan, fluoroquinolones and third-generation cephalosporins are commonly and frequently used for the treatment of bacterial infections in humans and their companion animals [4,5]. Consequently, fluoroquinolone-resistant and/or extended-spectrum β-lactamase (ESBL)-producing *Enterobacterales* have been isolated from both human and companion animals [6,7]. Previously, we reported that fluoroquinolone-resistant and/or ESBL-producing *E. coli* colonized the guts of companion animals [7,8,9]. This suggests that the transmission of companion animal-derived resistant *E. coli* isolates to humans could occur by direct contact between owners and their animals. *E. coli* is one of the principal pathogens for urinary tract infection in humans [10]. Thus, transmission of companion animal-derived antimicrobial-resistant *E. coli* might pose a potential risk to human health [11]. The international high-risk *E. coli* clone, sequence type (ST) 131, is the fluoroquinolone-resistant and/or ESBL-producing *E. coli* most frequently isolated from human infections [12,13]. Recent studies also reported isolation of *E. coli* ST131 from companion animals [14,15,16]. However, it is unclear whether direct transmission of *E. coli* ST131 occurs between humans and companion animals.

To evaluate potential hazards associated with companion animal-derived antimicrobial-resistant bacteria to human health, investigation of the prevalence of antimicrobial resistance in companion animal bacterial isolates is required, particularly in relation to commonly and frequently used antimicrobial agents such as fluoroquinolones and third-generation cephalosporins. In addition, molecular epidemiological analysis of antimicrobial-resistant bacteria isolated from companion animals and their owners will shed light on the role of these animals in transmission of such isolates to humans, *vice versa*.

In this study, *E. coli* were isolated from the rectums of dogs and cats and the prevalence of fluoroquinolone-resistant and/or ESBL-producing *E. coli* was investigated to understand their possessions, especially focused on *E. coli* ST131, in rectum of companion animals. In some cases, *E. coli* isolates from companion animals and their owners was compared to estimate the possibility of the domestic cross-transmission.

## 2. Results

### 2.1. Isolation of E. coli from Rectal Samples Taken from Companion Animals, and Measurement of Their Susceptibility to Ciprofloxacin and Cefotaxime

Using plain-CHROMagar ECC, ciprofloxacin-added CHROMagar ECC, and CHROMagar ECC with ESBL supplement, 101 *E. coli* isolates, 46 ciprofloxacin-resistant *E. coli* isolates, and 28 ESBL-producing *E. coli* isolates were detected from rectal samples taken from dogs and cats, respectively (Table 1).

The antimicrobial susceptibility testing showed a total of 27.7% (*n* = 28/101) ciprofloxacin-resistant and 24.8% (*n* = 25/101) cefotaxime-resistant *E. coli* isolates from plain-CHROMagar ECC, respectively. Using ciprofloxacin-added CHROMagar ECC, 46 ciprofloxacin-resistant isolates were obtained of which 35 (76.1%) isolates showed resistant to cefotaxime. In addition, CHROMagar ECC with ESBL supplement showed 28 cefotaxime-resistant-isolates of which 25 (89.3%) isolates were ciprofloxacin-resistant (Table 1). There were significant differences in prevalence of ciprofloxacin resistance and cefotaxime resistance between plain-CHROMagar ECC and ciprofloxacin-added CHROMagar ECC/CHROMagar ECC with ESBL supplement (*p* < 0.01).

For *E. coli* isolates obtained from the plain-CHROMagar ECC, the minimum inhibitory concentration (MIC) of ciprofloxacin ranged from <0.25 to >32 mg/L (the MIC_50_ and MIC_90_ were <0.25 and >32 mg/L, respectively). Cefotaxime MIC ranged from <0.25 to >128 mg/L (MIC_50_ and MIC_90_ were 0.5 and >128 mg/L, respectively). The prevalence of both of ciprofloxacin and cefotaxime resistance was 20.1% (21/101 isolates). For *E. coli* isolates obtained from the ciprofloxacin-added CHROMagar ECC, all isolates exhibited ciprofloxacin resistance, the ciprofloxacin MIC ranged from 8 to >32 mg/L (MIC_50_ and MIC_90_ were 32 and >32 mg/L, respectively). Cefotaxime MIC ranged from <0.25 to >128 mg/L (both the MIC_50_ and MIC_90_ were >128 mg/L). For *E. coli* isolates obtained from CHROMagar ECC with ESBL supplement, the ciprofloxacin MIC ranged from 0.5 to >32 mg/L (MIC_50_ and MIC_90_ were 32 and >32 mg/L, respectively). All isolates exhibited cefotaxime resistance, and the cefotaxime MIC ranged from 16 to >128 mg/L (both of MIC_50_ and MIC_90_ was >128 mg/L).

### 2.2. Isolation Rate of Ciprofloxacin- or Cefotaxime-Resistant E. coli Isolates from Companion Animals

The isolation rate of ciprofloxacin- and cefotaxime-resistant organisms among 34 companion animals using plain-CHROMagar ECC was 32.4% and 29.4%, respectively. Although the rates were mostly higher in ciprofloxacin-added CHROMagar ECC and CHROMagar ECC with ESBL supplement (Table 2), there were no significant differences between plain-CHROMagar ECC and ciprofloxacin-added CHROMagar ECC or CHROMagar ECC with ESBL supplement (*p* > 0.05).

### 2.3. Determination of ST131 Clades in Ciprofloxacin-Resistant E. coli Isolates

Of the *E. coli* isolates cultured on plain-CHROMagar ECC, 9.9% (10/101 isolates) were from the ST131 clade (Table 1). This accounted for 35.7% (10/28 isolates) of the ciprofloxacin-resistant *E. coli* isolated on this medium. The isolation rate of ST131 from companion animals was 11.8% (4/34 dogs and cats). Of the 10 ST131 isolates, seven (6.9% of total ciprofloxacin-resistant *E. coli* isolates) were from clade C1-M27 and three (3.0% of total ciprofloxacin-resistant *E. coli* isolates) were from clade C1-nM27.

Of the 46 ciprofloxacin-resistant *E. coli* isolates obtained from ciprofloxacin-added CHROMagar, 15 (32.6%) were ST131 isolates. The isolation rate of ST131 from companion animals was 14.7% (5/34 dogs and cats). Of the 15 ST131 isolates, nine (19.6% of total ciprofloxacin-resistant *E. coli* isolates) were from clade C1-M27 and six (13.0% of total ciprofloxacin-resistant *E. coli* isolates) were from clade C1-nM27. The prevalence of clade C1-nM27 isolates was significantly higher (*p* < 0.05) when isolates were grown on ciprofloxacin-added CHROMagar ECC than when grown on plain-CHROMagar ECC (Table 1).

Of the 28 ESBL-producing *E. coli* isolates from CHROMagar ECC with ESBL supplement, 10 (35.7%) were ST131 isolates. The isolation rate of ST131 from companion animals was 11.8% (4/34 dogs and cats). Of the 10 ST131 isolates, eight (28.6% of total ESBL-producing *E. coli* isolates) were from clade C1-M27 and two (7.1% of total ESBL-producing *E. coli* isolates) were from clade C1-nM27. The prevalence of clade C1-M27 isolates was significantly higher (*p* < 0.01) when grown on CHROMagar ECC with ESBL supplement than when grown on plain-CHROMagar ECC (Table 1).

### 2.4. Detection of CTX-M Type β-Lactamase

Multiplex polymerase chain reaction (PCR) detected a CTX-M type β-lactamase gene in 28 cefotaxime-resistant *E. coli* isolates obtained from CHROMagar ECC with ESBL supplement. Six isolates possessed a CTX-M group 1 β-lactamase gene, whereas 21 possessed a CTX-M group 9 β-lactamase gene. One strain possessed a β-lactamase gene that did not belong to CTX-M group 1, group 2, or group 9. All CTX-M group 1 β-lactamase genes were identified as CTX-M-15 by DNA Sanger sequencing. CTX-M group 9 genes were shown to be one of two types: CTX-M-14 (13 isolates) or CTX-M-27 (eight isolates).

### 2.5. Multi-Locus Sequence Typing (MLST) Analysis

STs of 16 ciprofloxacin-resistant *E. coli* isolates from individual dogs and cats were identified. These were ST131 (five isolates), ST10 (three isolates), ST1193 and ST2380 (each two isolates), and ST38, ST648, ST5150, and ST5163 (one isolate each).

### 2.6. MLST Analysis of Ciprofloxacin-Resistant E. coli Isolates from Companion Animals and Their Owners

Rectal swab samples were obtained from the owners of three dogs (#1, #2, and #3; Table 3). Ciprofloxacin-resistant *E. coli* isolates were obtained from dog #1 and its owner under all three culture conditions, and all were ST131 C1-M27 isolates. Ciprofloxacin-resistant *E. coli* isolates were isolated from dog #2, with both plain- and ciprofloxacin-added CHROMagar ECC, and from its owner with ciprofloxacin-added CHROMagar ECC. All isolates from dog #2 and its owner were ST131 C1-nM27. Ciprofloxacin-resistant *E. coli* isolates were isolated from dog #3 with both ciprofloxacin-added CHROMagar ECC and CHROMagar ECC with ESBL supplement, and both were ST2380 isolates. An ST2380 isolate was also recovered from one of the two owners using ciprofloxacin-added CHROMagar ECC.

## 3. Discussion

In this study, we isolated fluoroquinolone-resistant or ESBL-producing *E. coli* isolates from the rectums of dogs and cats in Japan using three different selective media to evaluate the potential prevalence of antimicrobial-resistant *E. coli* in companion animals.

The prevalence of ciprofloxacin- and cefotaxime-resistant *E. coli* isolated using plain-CHROMagar was 27.7% and 24.8%, respectively. These prevalence values were lower than the prevalence of similar antibiotic-resistant *E. coli* isolates from diseased dogs (38.8% for ciprofloxacin-resistant and 26.5% for cefotaxime-resistant) and cats (37.5% for ciprofloxacin-resistant and 26.6% for cefotaxime-resistant) reported in a national surveillance study carried out in Japan, Japanese Veterinary Antimicrobial Resistance Monitoring System (JVARM) [17]. The current study was based on swabs taken from animals attending companion animal clinics. As such, the data presented may reflect a selection bias because the samples came from dogs and cats without overt bacterial infections and which are less likely to have been previously exposed to antimicrobial treatments. On the other words, it indicates that companion animals harbour the ciprofloxacin- and cefotaxime-resistant *E. coli* in certain manner in their rectums. The prevalence of both ciprofloxacin and cefotaxime resistance was 20.1% in plain-CHROMagar, and the prevalence of ciprofloxacin- or cefotaxime-resistant *E. coli* isolates was significantly increased by the presence of ciprofloxacin and ESBL supplement in the selection medium (Table 1). This observation is very important because if the ciprofloxacin-resistant *E. coli* isolates were selected, these mostly exhibit resistant to another frequent-used antimicrobial agents, cephalosporins (ESBL-producing). The co-resistance of fluoroquinolone-resistance and the third-generation cephalosporins should be taken into account in clinical settings in order to optimize antimicrobial treatment efficacy. Most of the cefotaxime-resistant *E. coli* isolates possessed either CTX-M-14, CTX-M-15, or CTX-M-27 type β-lactamases. These CTX-M types are widespread in human clinical settings in Japan, and CTX-M-possessing isolates frequently show co-resistance to fluoroquinolones [18,19] as well as companion animals [7,20]. This suggests a common feature between humans and companion animals. Detection of CTX-M-14, CTX-M-15, or CTX-M-27 type β-lactamases genes has also been reported among dogs and cats from other Asian regions, America, Europe, and Oceania (only CTX-M-15 among the three types was found in Africa) [6]. Thus, spread of the CTX-M type β-lactamases should be international concerns not only in human [21] but also in companion animals.

In this study, the isolation rate of ciprofloxacin-resistant and cefotaxime-resistant *E. coli* isolates from companion animals was 32.4% and 29.4%, respectively. Although there were no significances, slightly higher isolation rates of resistant isolates were observed when using ciprofloxacin-added or ESBL supplement agar. In the latter case, ciprofloxacin-resistant or ESBL-producing *E. coli* isolates should be predominant in the presence of additional antimicrobial selection, as shown by the increase of ciprofloxacin- and cefotaxime-resistant *E. coli* isolates in the selective medium (Table 1). Consistent with this, a previous study demonstrated that administration of first-generation cephalosporins (cefazolin and cephalexin) to dogs selected ESBL-producing *E. coli* isolates in feces [22].

According to MLST analysis, we identified eight fluoroquinolone-resistant *E. coli* lineages (ST131, ST10, ST1193, ST2380, ST38, ST648, ST5150, and ST5163). The ST131, ST10, ST1193, ST38, and ST648 lineages constituted 75.0% (12/16) of the total fluoroquinolone-resistant isolates, and all have been isolated previously from human specimens [23,24]. This result suggests that fluoroquinolone-resistant *E. coli* are present in companion animals and, since these animals are in close contact with humans, they are a potential hazard to human health. The most isolated ST in this study was ST131 (31.3%; 5/16 isolates). ST131 is recognized as an international high-risk clone in human clinical settings [15]. Previous studies in Japan also reported isolation of ST131 from ESBL-producing *E. coli* isolates in dogs and cats, and that ST131 was the predominant lineage [16,20]. Studies on fluoroquinolone-resistant *E. coli* isolates in human clinical settings reported that ST131 lineages accounted for more than 50% of these isolates [18,25]. Collectively, these studies suggest that ST131 is the predominant lineage in humans, but that it may be maintained in both humans and companion animals.

The recently emerged ST1193 lineage is also a pandemic fluoroquinolone-resistant clone [24]. In our previous study, neither ST131 nor ST1193 lineages were isolated from companion animals in the year of isolation, 2005 [26]. Moreover, our recent study revealed that, although ST1193 was not isolated from the human clinical setting in Japan (Sapporo) in 2008, it was subsequently isolated in 2020 [19]. These observations suggest that the pattern of emergence of fluoroquinolone-resistant *E. coli* isolates matches the *E. coli* clones emerging in humans. Consistent with this, all of the paired samples between companion animals and their owners shared the same fluoroquinolone-resistant *E. coli* lineages, including ST131 (Table 3). Additionally notable was that the same *E. coli* lineage, ST2380, was isolated from dog #3 and one of its two owners. This result also supports the possibility of domestic direct transmission of fluoroquinolone-resistant *E. coli* between humans and companion animals. In the limitations, the current study was conducted by small numbers of canine samples and paired (dogs and their owners) samples from limited animal clinics. Thus, the expanded studies will be required in future to further verify the probability. Although recent studies favor the possibility of cross transmission of fluoroquinolone-resistant *E. coli* between human and dogs [27,28,29], evidence of the isolation of identical lineages between dogs and their owners have fewer reported. Only two cases of the isolation of identical lineages of ESBL-producing *E. coli* (ST131 and ST38) from both of human and the household dogs were reported [30]. Thus, current observations worth to strengthen the domestic transmission in dogs and their owners.

## 4. Materials and Methods

### 4.1. Bacterial Isolation

All 34 rectal samples obtained by rectal swab were collected from companion animals (28 dogs and 6 cats) that attended three animal clinics in Sapporo, Japan. These swabs were spread on CHROMagar ECC (Kanto Chemical, Tokyo, Japan). One to three blue colonies of suspected *E. coli* were picked, sub-cultured on CHROMagar ECC, and stored as 20% glycerol stocks at −80 °C until use for subsequent experiments.

For the selection of fluoroquinolone-resistant and ESBL-producing *E. coli* isolates, swabs were inoculated into 1 mL of Trypticase Soy Broth (Becton, Dickinson and Company, Franklin Lakes, NJ, USA) and incubated overnight at 37 °C. The resulting cultures were spread on ciprofloxacin (Bayer, Osaka, Japan) (1 mg/L)-added CHROMagar ECC and CHROMagar ECC supplemented with CHROMagar ESBL supplement (Kanto Chemical, Tokyo, Japan). Three colonies were picked up from each agar plate of all selective media. Identification of *E. coli* was performed using MALDI Biotyper (Bruker, Billerica, MA, USA). After receiving consent from the animal owners, *E. coli* from their feces was isolated as described above.

### 4.2. Antimicrobial Susceptibility

Susceptibility to ciprofloxacin and cefotaxime were determined by the broth microdilution method according to Clinical and Laboratory Standards Institute (CLSI) guidelines [31]. Isolates were inoculated into 1 mL of Trypticase Soy Broth and cultured overnight at 37 °C. After the cultivation, the suspensions were diluted and 2 μL of the dilutions (final inoculation was 5 × 10^5^ cfu/mL) were added into each well of 96 well plate containing ciprofloxacin or cefotaxime in 100 μL of cation-adjusted Mueller-Hinton broth (Becton, Dickinson and Company, Franklin Lakes, NJ, USA). MICs were determined after incubation for 20 h at 37 °C. Isolates that exceeded breakpoints (ciprofloxacin was 2 mg/L, and cefotaxime was 4 mg/L) were identified as resistant. *E. coli* ATCC 25922 was used as the reference strain.

### 4.3. Genetic Analysis

ST131, their clades (A, B, and C) and C subclades (C1-M27, C1-nM27, and C2) were identified by multiplex PCR, as described by Matsumura et al. [32]. Other STs of *E. coli* isolates were identified by MLST analysis as previously described by using primer sets of seven housekeeping genes (*adk*, *fumC*, *icd*, *purA*, *gyrB*, *recA*, and *mdh*) [33]. β-Lactamase gene CTX-M-1, 2, and 9 groups were detected by the multiplex PCR described by Dallenne et al. [34], and their variants were identified by Sanger DNA sequencing as described in a previous study [35]. All PCRs were performed by using Quick Taq HS Dye Mix (TOYOBO, Osaka, Japan) and MiniAmp Thermal Cycler (Thermo Fisher Scientific, Waltham, MA, USA) according to the same reaction conditions described in above references.

### 4.4. Statistical Analysis

Fisher’s exact test was used for the statistical analysis. A *p*-value of <0.05 was considered statistically significant.

### 4.5. Ethical Approvemnet

This study was approved by the Ethical committee of Sapporo Medical University (No. 1-2-23).

## 5. Conclusions

This study demonstrates that companion animals possess fluoroquinolone-resistant and ESBL-producing *E. coli* isolates in their rectums, the majority of which comprise the same fluoroquinolone-resistant lineages (ST131, ST10, ST1193, ST38, and ST648) detected previously in human clinical samples. In addition, we found that most fluoroquinolone-resistant *E. coli* isolates exhibited co-resistance to cefotaxime due to the presence of CTX-M type β-lactamase genes (CTX-M-14, CTX-M-15, and CTX-M-27). The increased prevalence of fluoroquinolone-resistant and/or ESBL-producing *E. coli* isolates when ciprofloxacin or ESBL supplement was included in the selective medium, suggests the need for prudence in the use and administration of antimicrobials in veterinary settings to prevent the selection of these resistant *E. coli*. Furthermore, it is also important for dog and cat owners to implement infection control measures in their homes to prevent domestic transmission of fluoroquinolone-resistant and/or ESBL-producing *E. coli* between humans and companion animals.

## Figures and Tables

**Table 1 antibiotics-10-01463-t001:** Prevalence of ciprofloxacin and cefotaxime resistance and ST131 clades in *E. coli* isolates derived from companion animals.

CHROMagar ECC	No. Total Isolates	Prevalence of Resistance (No. of Isolates)		Prevalence of ST131 (No. of Isolates)		
		Ciprofloxacin	Cefotaxime	Total	C1-M27	C1-nM27
Plain	101	27.7% (28)	24.8% (25)	9.9% (10)	6.9% (7)	3.0% (3)
Ciprofloxacin-added	46	100% (46) **	76.1% (35) **	32.6% (15) **	19.6% (9)	13.0% (6) *
ESBL supplement	28	89.3% (25) **	100% (28) **	35.7% (10) **	28.6% (8) **	7.1% (2)

**, *p* < 0.01; *, *p* < 0.05 vs. plain-CHROMagar ECC.

**Table 2 antibiotics-10-01463-t002:** Isolation rate of ciprofloxacin- and cefotaxime-resistant *E. coli* isolates among 34 companion animals.

CHROMagar ECC	Isolation Rate of Resistance (No. of Animals)	
	Ciprofloxacin	Cefotaxime
Plain	32.4% (11)	29.4% (10)
Ciprofloxacin-added	47.1% (16)	38.2% (13)
ESBL supplement	29.4% (10)	32.4% (11)

**Table 3 antibiotics-10-01463-t003:** Characteristics of *E. coli* isolates obtained from dogs and their owners.

*E. coli* Strains	Origin	CHROMagar ECC	MIC (mg/L)		MLST-Clade	*bla* _CTX-M_
			CIP	CTX		
DE3-1	Dog #1	plain	32	>128	ST131 C1-M27	CTX-M-27
DE3-2	Dog #1	plain	32	>128	ST131 C1-M27	CTX-M-27
DE3-3	Dog #1	plain	32	>128	ST131 C1-M27	CTX-M-27
DE3CIP1	Dog #1	Ciprofloxacin	32	>128	ST131 C1-M27	CTX-M-27
DE3CIP2	Dog #1	Ciprofloxacin	32	>128	ST131 C1-M27	CTX-M-27
DE3CIP3	Dog #1	Ciprofloxacin	32	>128	ST131 C1-M27	CTX-M-27
DE3ES1	Dog #1	ESBL	32	>128	ST131 C1-M27	CTX-M-27
DE3ES2	Dog #1	ESBL	32	>128	ST131 C1-M27	CTX-M-27
DE3ES3	Dog #1	ESBL	32	>128	ST131 C1-M27	CTX-M-27
DE3h1	Owner of dog #1	plain	32	>128	ST131 C1-M27	CTX-M-27
DE3h2	Owner of dog #1	plain	32	>128	ST131 C1-M27	CTX-M-27
DE3h3	Owner of dog #1	plain	32	>128	ST131 C1-M27	CTX-M-27
DE3hCIP1	Owner of dog #1	Ciprofloxacin	32	>128	ST131 C1-M27	CTX-M-27
DE3hCIP2	Owner of dog #1	Ciprofloxacin	32	>128	ST131 C1-M27	CTX-M-27
DE3hCIP3	Owner of dog #1	Ciprofloxacin	32	>128	ST131 C1-M27	CTX-M-27
DE3hES1	Owner of dog #1	ESBL	32	>128	ST131 C1-M27	CTX-M-27
DE3hES2	Owner of dog #1	ESBL	32	>128	ST131 C1-M27	CTX-M-27
DE3hES3	Owner of dog #1	ESBL	32	>128	ST131 C1-M27	CTX-M-27
FE1-1	Dog #2	plain	32	<0.25	ST131 C1-nM27	ND
FE1-2	Dog #2	plain	32	<0.25	ST131 C1-nM27	ND
FE1-3	Dog #2	plain	32	<0.25	ST131 C1-nM27	ND
FE1CIP1	Dog #2	Ciprofloxacin	32	<0.25	ST131 C1-nM27	ND
FE1CIP2	Dog #2	Ciprofloxacin	32	<0.25	ST131 C1-nM27	ND
FE1CIP3	Dog #2	Ciprofloxacin	32	<0.25	ST131 C1-nM27	ND
FE1hCIP1	Owner of dog #2	Ciprofloxacin	32	<0.25	ST131 C1-nM27	ND
FE1hCIP2	Owner of dog #2	Ciprofloxacin	32	<0.25	ST131 C1-nM27	ND
FE1hCIP3	Owner of dog #2	Ciprofloxacin	32	<0.25	ST131 C1-nM27	ND
GE1CIP1	Dog #3	Ciprofloxacin	32	>128	ST2380	CTX-M-14
GE1ES1	Dog #3	ESBL	32	>128	ST2380	CTX-M-14
GE1h-1	Owner A of dog #3	Ciprofloxacin	32	<0.25	ST8671	ND
GE1h-2	Owner B of dog #3	Ciprofloxacin	32	<0.25	ST2380	ND

CIP, ciprofloxacin; CTX, cefotaxime; ND, not detected.

## Data Availability

Data is contained within the article.

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
