# Peer review of "Isolation of Human Lineage, Fluoroquinolone-Resistant and Extended-β-Lactamase-Producing Escherichia coli Isolates from Companion Animals in Japan"

_antibiotics, 2021, doi:10.3390/antibiotics10121463_

Round 1

Reviewer 1 Report

The work is clearly presented and has great relevance for monitoring antimicrobial resistance; however, the authors should clarify some issues:
1.- It is not possible to talk about prevalence since they use a very small sample number not representative of any population the concept prevalence should be changed by positivity.
2.- It is not possible to mention the association between resistance in humans and animals, since a more robust epidemiological study should be conducted for this purpose. 

Author Response

Reviewer#1

The work is clearly presented and has great relevance for monitoring antimicrobial resistance; however, the authors should clarify some issues:

1.- It is not possible to talk about prevalence since they use a very small sample number not representative of any population the concept prevalence should be changed by positivity.

Reply; Thank you for your comment. As the reviewer mentioned, limited samples (n =34) were conducted in this study (we have added it as limitation in Discussion; line 217-219). Even so, this number was sufficient for statical analysis using Fisher's exact test.

2.- It is not possible to mention the association between resistance in humans and animals, since a more robust epidemiological study should be conducted for this purpose.

Reply; Thank you for your comment. Although the small sample sets (three pairs) of dogs and human (dogs’ owners) were conducted in this study, we have considered that these observations in this study are inevitable to discuss the association, because there are only a few previous reports that isolates with identical sequence type obtained from both of dogs and their owners in fluoroquinolone-resistant E. coli. We have added the limitations of current study (small sample numbers) and requirement of the expanding for our future study (lines 217-219). In addition, we have added the previous studies that support possible transmission of fluoroquinolone-resistant E. coli between human and dogs (line 220-225).

Reviewer 2 Report

Major concerns

1- The number of samples is very low. It is less likely to put a conclusion based on the analysis of 34 samples

2- You have 34 samples only and from each sample, you isolated more than one colony to have at the end 101 samples, based on this, you construct all your results. This is not correct, you have to consider using only one single colony for each sample

3- No stats included

Other comments

1- You need to provide the Ethical statement including the protocol number in a separate section

2- Please provide more details about the Antimicrobial susceptibility test

3- please provide the aim of the study at the end of the introduction

4- Please provide more details about the genetic analysis

5- Please provide a statistical analysis section in materials and methods

Author Response

Major concerns

1- The number of samples is very low. It is less likely to put a conclusion based on the analysis of 34 samples

Reply; Thank you for your comment. Although only limited samples (n =34) were conducted in this study, this number was sufficient for statistical analysis using Fisher's exact test.

2- You have 34 samples only and from each sample, you isolated more than one colony to have at the end 101 samples, based on this, you construct all your results. This is not correct, you have to consider using only one single colony for each sample

Reply; Thank you for your comment. We have considered that isolation of multiple colonies from one sample provided us more detailed and valuable information on colonization of E. coli in large intestine of companion animals than isolation of a single colony from one sample.

3- No stats included

Reply; I have added the descriptions about statistical analysis in Material and Methods (line 278-280)

Other comments

1- You need to provide the Ethical statement including the protocol number in a separate section

Reply; We have mentioned it in a separate section (line 282-284)

2- Please provide more details about the Antimicrobial susceptibility test

Reply; We have added it into the revised manuscript (line 260-266).

3- please provide the aim of the study at the end of the introduction

Reply; We have added it into the revised manuscript (line 63-66).

4- Please provide more details about the genetic analysis

Reply; We have added it into the revised manuscript (line 270-276).

5- Please provide a statistical analysis section in materials and methods

Reply; We have added it into the revised manuscript (line 278-280).

Reviewer 3 Report

  • Line 236, please indicate that you select one to three colonies from each selective agar. Or you select this (1-3 colonies) from both agar?
  • Line 77, please provide the number if isolates (collected from antibiotic-free medium) that are concomitantly resistant to ciprofloxacin and cefotaxime. Also in your discussion try to highlight that ciprofloxacin-resistance and ESBL production are mainly linked in the collected isolates.
  • If possible under the table 1 indicate the phenotype of ST131 isolates. For example among the 10 isolates collected from ‘Plain’ how many are susceptible to both antibiotic, how many resistant only to ciprofloxacin, how many are resistant to cefotaxime, how many are resistant to both antibiotics. OR, you can also few lines about this in your text.
  • In Table 3 , add the detected blaCTX-M gene
  • Line 189, delete the ‘This result suggests that’ since there is a repetition (This result suggests that This result suggests that fluoroquinolo…).

Author Response

Line 236, please indicate that you select one to three colonies from each selective agar. Or you select this (1-3 colonies) from both agar?

Reply; Number of colonies picked up was three per each selective agar, thus we have corrected the text (line 252).

Line 77, please provide the number if isolates (collected from antibiotic-free medium) that are concomitantly resistant to ciprofloxacin and cefotaxime. Also in your discussion try to highlight that ciprofloxacin-resistance and ESBL production are mainly linked in the collected isolates.

Reply; The prevalence of resistance to both ciprofloxacin and cefotaxime was 20.1% (21/101 isolates). We have added it in Results of our revised manuscript (line 85-86). In addition, we have added the descriptions about co-resistance to ciprofloxacin and cefotaxime in Discussion (line 170-182).

If possible under the table 1 indicate the phenotype of ST131 isolates. For example among the 10 isolates collected from ‘Plain’ how many are susceptible to both antibiotic, how many resistant only to ciprofloxacin, how many are resistant to cefotaxime, how many are resistant to both antibiotics. OR, you can also few lines about this in your text.

Reply; We have added the descriptions in the text (line 70-83 and 87-88).

In Table 3 , add the detected blaCTX-M gene

Reply; We have added blaCTX-M types in Table 3.

Line 189, delete the ‘This result suggests that’ since there is a repetition (This result suggests that This result suggests that fluoroquinolo…).

Reply; Thank you for your indication. I have removed it.

Reviewer 4 Report

The current study investigated the   Human Lineage, Fluoroquinolone-Resistant and   Extended-β-Lactamase-Producing Escherichia coli Isolates from   Companion Animals in Japan. This study is interesting epidemiological study. However, there are several issues need to be corrected, resolved, or clarified to improve the contents of the manuscript. I recommend the authors to major revise their manuscript as follow:

  • All words in abstract and main text of manuscript (such as PCR) need to be written in full term in first mention. Then their abbreviations are written. Correct them through manuscript.
  • Results, page 2 and 3, lines 65-100: the results are written in confusing manner. I suggest to rewrite this section and give a total and percentage of islates with each methods.
  • Results, page 2, line 67-69: “briefly………” need to be written in better manner such as: “Using plain-CHROMagar ECC, ciprofloxacin-added CHROMagar ECC, and CHROMagar ECC with ESBL supplement, 101 E. coli, 46  ciprofloxacin-resistant E. coli, and  28 ESBL producing E. coli were detected from rectal samples taken from dogs and cats, respectively (Table 1).
  • Results, page 2, line 69: the authors claimed that “There were no significant differences in the isolation rates between dogs and cats (Table 2)”. In table 2, no data are mentioned regards dogs and cats. Also, there is no any p-value to reveal absence of significant association claimed by authors.
  • No data about the number of cat and dogs are mentioned in the manuscript in result section.
  • Discussion: this part need more work as most of the contents are repeat of the results. The authors need to compare their results with previous studies in this regards. Also, the strengths and limitations of the study need to be clarified.
  • Materials and methods: No quality controls strains are mentioned in this section. Also, the company name and their countries need to be written for all materials used such as antibiotic powders.
  • References: Several references are not update and are before 2016. I suggest to update and replaced them with more recent references.
  • I recommend to use below references in your discussion as recent references:
  • Ejikeugwu C, Nworie O, Saki M, Al-Dahmoshi HO, Al-Khafaji NS, Ezeador C, Nwakaeze E, Eze P, Oni E, Obi C, Iroha I. Metallo-β-lactamase and AmpC genes in Escherichia coli, Klebsiella pneumoniae, and Pseudomonas aeruginosa isolates from abattoir and poultry origin in Nigeria. BMC microbiology. 2021 Dec;21(1):1-9.
  • Nzima B, Adegoke AA, Ofon UA, Al-Dahmoshi HO, Saki M, Ndubuisi-Nnaji UU, Inyang CU. Resistotyping and extended-spectrum beta-lactamase genes among Escherichia coli from wastewater treatment plants and recipient surface water for reuse in South Africa. New Microbes and New Infections. 2020 Nov 1;38:100803.

Author Response

The current study investigated the Human Lineage, Fluoroquinolone-Resistant and   Extended-β-Lactamase-Producing Escherichia coli Isolates from Companion Animals in Japan. This study is interesting epidemiological study. However, there are several issues need to be corrected, resolved, or clarified to improve the contents of the manuscript. I recommend the authors to major revise their manuscript as follow:

All words in abstract and main text of manuscript (such as PCR) need to be written in full term in first mention. Then their abbreviations are written. Correct them through manuscript.

Reply; Thank you for your suggestion. We have written PCR as polymerase chain reaction in Abstract, and confirmed the use of abbreviations throughout the manuscript (line 18-19).

Results, page 2 and 3, lines 65-100: the results are written in confusing manner. I suggest to rewrite this section and give a total and percentage of islates with each methods.

Reply; Thank you for your suggestion. We have corrected this section as to be total number of isolates in the first paragraph, prevalence of ciprofloxacin and cefotaxime resistance in the second paragraph, and MIC ranges in the third paragraph (line 70-83)

Results, page 2, line 67-69: “briefly………” need to be written in better manner such as: “Using plain-CHROMagar ECC, ciprofloxacin-added CHROMagar ECC, and CHROMagar ECC with ESBL supplement, 101 E. coli, 46 ciprofloxacin-resistant E. coli, and 28 ESBL producing E. coli were detected from rectal samples taken from dogs and cats, respectively (Table 1).

Reply; Thank you for your suggestion. We have corrected this sentence as the reviewer’s suggestion (line 70-73).

Results, page 2, line 69: the authors claimed that “There were no significant differences in the isolation rates between dogs and cats (Table 2)”. In table 2, no data are mentioned regards dogs and cats. Also, there is no any p-value to reveal absence of significant association claimed by authors. No data about the number of cat and dogs are mentioned in the manuscript in result section.

Reply; Thank you for your suggestions. After confirmation, we have removed the descriptions about comparison between dogs and cats, because number of cats conducted in this study (n= 6) was small and it was not suitable for statistical analysis.

Discussion: this part need more work as most of the contents are repeat of the results. The authors need to compare their results with previous studies in this regards. Also, the strengths and limitations of the study need to be clarified.

Reply; Thank you for your suggestions. We have added the previous studies (line 181, 220-225) and discussed them. In addition, we have added limitations of this study (line 217-218).

Materials and methods: No quality controls strains are mentioned in this section. Also, the company name and their countries need to be written for all materials used such as antibiotic powders.

Reply; The reference strain (ATCC25922) in Antimicrobial susceptibility has added (line 266). Re-checked product information was properly described.

References: Several references are not update and are before 2016. I suggest to update and replaced them with more recent references.

I recommend to use below references in your discussion as recent references:

Ejikeugwu C, Nworie O, Saki M, Al-Dahmoshi HO, Al-Khafaji NS, Ezeador C, Nwakaeze E, Eze P, Oni E, Obi C, Iroha I. Metallo-β-lactamase and AmpC genes in Escherichia coli, Klebsiella pneumoniae, and Pseudomonas aeruginosa isolates from abattoir and poultry origin in Nigeria. BMC microbiology. 2021 Dec;21(1):1-9.

Nzima B, Adegoke AA, Ofon UA, Al-Dahmoshi HO, Saki M, Ndubuisi-Nnaji UU, Inyang CU. Resistotyping and extended-spectrum beta-lactamase genes among Escherichia coli from wastewater treatment plants and recipient surface water for reuse in South Africa. New Microbes and New Infections. 2020 Nov 1;38:100803.

Reply; Thank you for your recommendation. I have added the latest references that were investigated fluoroquinolone-resistant and/or ESBL-producing E. coli from companion animals [ref 20, 21, 27-30]. The references recommended by the reviewer are not data of companion animals, and I consider that these are less related to the aim of our present study.

Round 2

Reviewer 2 Report

Thank you for addressing my comments

Author Response

Comments and Suggestions for Authors Thank you for addressing my comments

Reply; Thank you very much for you comments.

Reviewer 4 Report

In the Results section, page 2, line 74-81, the comparison among the three different media is not corrected, as each medium has its own properties. Thus rewrite the “Prevalence of ciprofloxacin  resistance  from  plain-CHROMagar  ECC, ciprofloxacin- added  CHROMagar  ECC,  and  CHROMagar  ECC  with  ESBL  supplement  were  27.7%, 100%, and 89.3%, respectively (Table 1).  Prevalence of cefotaxime resistance from plain-CHROMagar ECC, ciprofloxacin-added CHROMagar ECC, and CHROMagar ECC with ESBL supplement were 24.8%, 76.1%, and 100%, respectively (Table 1). There were significant  differences  in  prevalence  of  ciprofloxacin  resistance  and  cefotaxime resistance  between plain-CHROMagar ECC and ciprofloxacin-added CHROMagar ECC/CHROMagar ECC with ESBL supplement (p <0.01).” rewrite this section as follows:

“ The antibiotic susceptibility testing showed a total of 27.7% (n = 28/101) ciprofloxacin-resistant and 24.8% (n = 25/101) cefotaxime-resistant E. coli isolates, respectively. Using ciprofloxacin-added CHROMagar ECC, 46 ciprofloxacin-resistant isolates were obtained of which 35 (76.1%) isolates showed resistance to cefotaxime. Also, CHROMagar ECC with ESBL supplement showed 28 cefotaxime-resistant isolates of which 25 (89.3%) isolates were ciprofloxacin-resistant.”

Also delete the p=value and all asterisk signs from table 1. Delete all such comparison throughout manuscript.

Discussion need more works. The importance of beta lactamases producing gram negative bacteria especially CTX-M-14, CTX-M-15 ESBLs need to be clarified and the results need to be compared with other regions.

I suggest to cite the following references:

Ejikeugwu C, Nworie O, Saki M, Al-Dahmoshi HO, Al-Khafaji NS, Ezeador C, Nwakaeze E, Eze P, Oni E, Obi C, Iroha I. Metallo-β-lactamase and AmpC genes in Escherichia coli, Klebsiella pneumoniae, and Pseudomonas aeruginosa isolates from abattoir and poultry origin in Nigeria. BMC microbiology. 2021 Dec;21(1):1-9.

Ejikeugwu C, Hasson SO, Al-Mosawi RM, Alkhudhairy MK, Saki M, Ezeador C, Eze P, Ugwu M, Duru C, Ujam NT, Edeh C. Occurrence of FOX AmpC gene among Pseudomonas aeruginosa isolates in abattoir samples from south-eastern Nigeria. Reviews in Medical Microbiology. 2020 Apr 1;31(2):99-103.

Author Response

Reviewer#4

In the Results section, page 2, line 74-81, the comparison among the three different media is not corrected, as each medium has its own properties. Thus rewrite the “Prevalence of ciprofloxacin  resistance  from  plain-CHROMagar  ECC, ciprofloxacin- added  CHROMagar  ECC,  and  CHROMagar  ECC  with  ESBL  supplement  were  27.7%, 100%, and 89.3%, respectively (Table 1).  Prevalence of cefotaxime resistance from plain-CHROMagar ECC, ciprofloxacin-added CHROMagar ECC, and CHROMagar ECC with ESBL supplement were 24.8%, 76.1%, and 100%, respectively (Table 1). There were significant  differences  in  prevalence  of  ciprofloxacin  resistance  and  cefotaxime resistance  between plain-CHROMagar ECC and ciprofloxacin-added CHROMagar ECC/CHROMagar ECC with ESBL supplement (p <0.01).” rewrite this section as follows:

“ The antibiotic susceptibility testing showed a total of 27.7% (n = 28/101) ciprofloxacin-resistant and 24.8% (n = 25/101) cefotaxime-resistant E. coli isolates, respectively. Using ciprofloxacin-added CHROMagar ECC, 46 ciprofloxacin-resistant isolates were obtained of which 35 (76.1%) isolates showed resistance to cefotaxime. Also, CHROMagar ECC with ESBL supplement showed 28 cefotaxime-resistant isolates of which 25 (89.3%) isolates were ciprofloxacin-resistant.”

Reply; Thank you for your suggestions. We had corrected these sentences as the reviewer’s mentioned (line 74-79).

Also delete the p=value and all asterisk signs from table 1. Delete all such comparison throughout manuscript.

Reply; We disagree with that suggestion because we have observed significant differences by Fisher's exact test.

Discussion need more works. The importance of beta lactamases producing gram negative bacteria especially CTX-M-14, CTX-M-15 ESBLs need to be clarified and the results need to be compared with other regions.

Reply; We have added the discussion that described about CTX-M type in other region and the importance of worldwide spread of CTX-M-14, CTX-M-15, and CTX-M-27, should be international concerns not only for human but also for companion animals (line 183-188).

I suggest to cite the following references:

Ejikeugwu C, Nworie O, Saki M, Al-Dahmoshi HO, Al-Khafaji NS, Ezeador C, Nwakaeze E, Eze P, Oni E, Obi C, Iroha I. Metallo-β-lactamase and AmpC genes in Escherichia coli, Klebsiella pneumoniae, and Pseudomonas aeruginosa isolates from abattoir and poultry origin in Nigeria. BMC microbiology. 2021 Dec;21(1):1-9.

Ejikeugwu C, Hasson SO, Al-Mosawi RM, Alkhudhairy MK, Saki M, Ezeador C, Eze P, Ugwu M, Duru C, Ujam NT, Edeh C. Occurrence of FOX AmpC gene among Pseudomonas aeruginosa isolates in abattoir samples from south-eastern Nigeria. Reviews in Medical Microbiology. 2020 Apr 1;31(2):99-103.

Reply; The references recommended by the reviewer are not data of companion animals, and I consider that these are less related to the aim of our present study.